# Sustainable Waste Logistics and the Development of Trade in Recyclable Raw Materials in Poland and Hungary

**Agata Mesjasz-Lech** [1,*] **and Pál Michelberger** [2]

[1]  Faculty of Management, Czestochowa University of Technology, 42-201 Częstochowa, Poland
[2]  Donát Bánki Faculty of Mechanical and Safety Engineering, Óbuda University, 1081 Budapest, Hungary
*  Correspondence: agata.mesjasz-lech@wz.pcz.pl; Tel.: +48-34-3250-388

**Abstract:** This article aims to propose a methodological framework to determine the degree of the dynamic impact of the effect of activities in the field of sustainable waste logistics on the development of trade in recyclable raw materials in the chosen countries of the European Union, especially in Poland and Hungary. In order to determine the dynamic interdependence between the indicated phenomena, econometric tools associated with the vector autoregression model were used, namely: Granger causality tests, impulse response function and variance decomposition of forecast errors. The tools used will not only identify the direction of the causal interdependence between the effects of sustainable logistics activities and the development of trade in recyclable raw materials, but also allow to determine the strength of the interaction between these variables. The conducted research shows that changes in the environmental effects of waste logistics activities are the Granger cause of changes occurring in the trade of renewable raw materials, especially in Hungary. Considering adequately delayed values of the synthetic development measure of the environmental effects of waste logistics increases the accuracy of predictions for changes in the trade of recyclable raw materials.

**Keywords:** recyclable raw materials management; sustainability; waste logistics; waste management

## 1. Introduction

A European Citizen produced 487 kg municipal waste on average in 2017—only 29% of this amount was recycled only [1]. Just over the 40% of the solid waste is reused or recycled in EU. In several member states this ratio exceeds 70%, but there are also EU countries where 75% of municipal waste is landfilled (Figure 1). There are undoubtedly opportunities for progress in this area. Good waste management methods and practices can be found in the literature [2,3].

The European Union repeatedly formulated aims, plans and recommendations concerning waste management [4–14]. A common EU aim is to recycle 65% of municipal waste and 75% of packaging waste by 2030 [15]. The document of "General Union Environment Action Programme to 2020; Living well, within the limits of our planet" described a waste management hierarchy according to environmental aspects [16]:

- prevention (avoid the amount of waste);
- reduce of waste;
- preparing for re-use (e.g., selective waste collection);
- recycling and waste treatment;
- other recovery, and disposal.

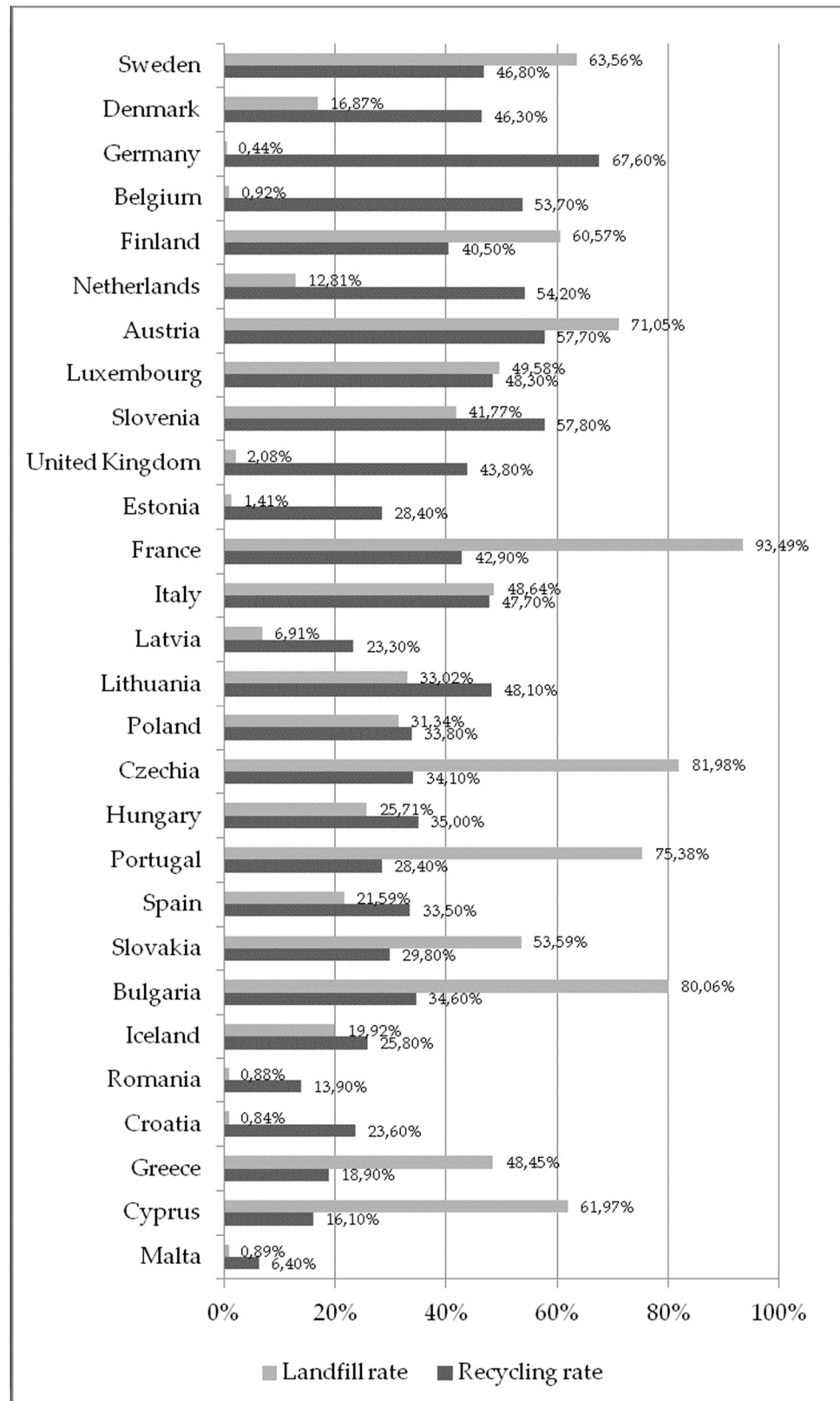

**Figure 1.** Landfill and recycling rates for municipal waste in European Union countries in 2017 (in %).

The prevention of waste formation and the reduction of waste are considered more important than recycling and waste treatment in the European Union. The least desirable method of waste disposal is to landfill. The goal of waste management is slowly changing in the European Union.

It can be observed that there is a shift towards a more holistic approach in the analysis of waste management [17–19], and reducing environmental impact is the priority for future generations. Waste minimisation techniques should be implemented as well, considering the principles of sustainable development [20–26]. Importantly, implementing sustainable development does not cause long-term business disadvantages for the companies [27]. Many European cities have been using sustainable methods in waste management for years, keeping the amount of generated and collected mixed waste to a minimum. Unfortunately, both in Poland and Hungary, the dominant method of waste disposal is landfilling (Figure 2). Waste minimisation techniques are applicable in waste reduction and in the rationalizaton of municipal waste treatment [28–33]. Besides recycling and source reduction, there is the possibility to modify the products to minimise waste (Figure 2). The aim of zero waste strategy is to organise such business and production processes, where the waste outputs—preferably completely—become input in other processes.

**Figure 2.** Possibilities of waste minimization [34].

The pro-environmental orientation of organizations, which leads to understanding environmental protection as one of their objectives, can be the basis for creating new markets or reorganizing existing ones from the point of view of specific material flows in economic systems. The problem of waste management in the European Union is mainly caused by [35–38]:

- the increase in industrialization and urbanization;
- the increase in the amount of waste generated per capita;
- the need to maintain a high level of investment into infrastructure (incinerators, landfills, recycling facilities);
- institutional barriers;
- the diversity of interest groups;
- political and legal changes in the field of waste management.

What seems to be an important problem is the creation of all kinds of legal regulations in relation to entities dealing with waste treatment and entities using secondary raw materials. Yet, current legal acts mainly address methods of safe disposal of waste without determining the fraction of waste that should return to the economic system. Lack of solutions in this respect may stem from seeing pro-environmental activities as an obligation resulting from the ongoing degradation of the environment, and not as a way of achieving the profitability and productivity goals of enterprises.

## 2. Development of Sustainable Waste Logistics Concept Results in Trade of Recyclable Raw Materials

The dynamic development of methods and management concepts led to the implementation of environmental activities in the activities of organizations and fostered the development of tools

for their realization. This process resulted in waste logistics, which is an environmentally oriented concept. The basic goal is to shape the flow of materials and products in the opposite direction to the classic approach in order to minimize the amount of waste and to make the best use of waste suitable for recovery. Logistics concepts understood in this way supports sustainable activities related to the flow of materials [39–47]. Three basic logistic processes can be distinguished in waste management: gathering, collection and transport of waste. The concept of waste gathering covers every activity that prepares waste for transport to recovery or disposal sites. Gathering is therefore the first logistics process of waste management. This step takes place right after the waste generation stage—usually close to the place where it was generated—and the waste producer itself is the subject of the process. It is important to distinguish waste collection from transport: the collection consists of reception of waste from the producers; by contrast, the concept of waste transport is understood as the processes from the moment of completion of collection to their transfer to the place of processing. Transport is a very important element of the solid waste management due to the costs associated with it and the impact on the natural environment. The organization of transport, including the choice of means of transport, route optimization depends on the environmental, social and economic factors of a given country or even the region. Thus, it is likely to vary considerably between countries.

Solutions provided by waste logistics can help establish closed-loop material flows in economic systems [48]. In closed-loop material flows the concept of waste goes beyond its typical definition. They are no longer a useless reminder of processes carried out in an enterprise, but a valuable resource that can find its way back to the economic system [49]. Creating truly closed economic systems will contribute mainly to [50–53]:

− decrease in the level of environmental pollution;
− reduction in the use of natural resources;
− reduction in the capital and energy intensity in the processes of obtaining and processing secondary raw materials;
− respecting the rules of sustainable development.

In this context the analysis of the relationships between the effects of activities undertaken in the framework of waste logistics and the development of trade in recyclable raw materials appears to be justified. The research in this respect so far has concentrated on:

− the optimization of flows in closed-loop supply chains [54–57];
− managing secondary raw materials with the use of the transfer function [58–60];
− managing reverse logistics with the use of metaheuristic algorithms [61];
− investigation the dynamics of a closed-loop systems with autoregressive demand and return processes [62].

The existing literature on the subject lacs research on the dynamic influence of the trade of renewable raw materials on the pro-ecological effects. Hence, the goal of the article is to propose a methodological framework to determine the degree of the dynamic impact of the effect of activities in the field of sustainable waste logistics on the development of trade in recyclable raw materials. The following hypothesis was formulated:

> The influence between effect of activities in the field of sustainable waste logistics and the development of trade in recyclable raw materials in countries of Central Eastern Europe can be described with the vector autoregressive model (VAR).

It was assumed that the formation of closed economic loops through the implementation of waste logistics solutions translates into the development of renewable raw materials. This is expected to result in pro-ecological effects such as the reduction of waste, limited use of water and energy and the reduction of air pollution.

### 3. Materials and Methods

In order to determine the dynamic relationship between the effects of activities undertaken in the framework of waste logistics and the development of trade in recyclable raw materials, models of multidimensional stochastic processes were used—particularly the vector autoregressive model (VAR). Econometric modelling of multidimensional time series can be characterized by three basic principles:

1.　no distinction is made between endogenous and exogenous variables;
2.　the values of the model parameters are not limited;
3.　at the basics of the modelling process there is no strict and primary economic theory.

In addition, the study was supplemented by the Granger causality analysis, the determination of the impulse response function and the decomposition of the forecast error variance. A detailed description of the used methods can be found in Maddala [63] and Osińska [64].

The following variables were analyzed:

1.　Balance: the difference between exports and imports of waste and by-products (in tonne) in the trade of renewable raw materials. This variable measures the number of categories of waste and by-products moved within the borders of the European Union. It was also assumed that the analysis would cover the data on exports to and imports from countries which are not members of the European Union.
2.　Measure of development (MR): a synthetic variable made up of both positive and negative effects of pro-ecological activities undertaken by various entities (or failure to undertake them) in the area of waste logistics. The measure was estimated with the use of the development pattern method, classified in the group of linear ordering methods in multidimensional statistical analysis. The effects of pro-ecological activities for the countries of the European Union were described with the following set of variables:

- Final energy consumption (million tonnes of oil equivalent per capita);
- Domestic material consumption (tonnes per capita);
- Sulphur oxides (tonnes per capita);
- Nitrogen oxides (tonnes per capita);
- Waste generation (tonnes per capita).

The variables constituting the measure of development reflect the effects of individual processes in the field of waste logistics—in particular collection, transport and landfill of waste.

To estimate synthetic variable the following measures were counted in the next steps:

(1)　Unitarized variables (Equation (1)):

$$z_{ij} = \frac{x_{ij} - \overline{x}_j}{s_j}, \; (i = 1, 2, \ldots, n; \; j = 1, 2, \ldots, m), \tag{1}$$

where $n$ is the number of countries, $m$ is the number of variables, $z_{ij}$ is the standardized value of variable $X_j$, $\overline{x}_j$ is the arithmetic average of variable $X_j$ and $s_j$ is the standard deviation of variable $X_j$.

(2)　Pattern and anti-pattern (Equations (2) and (3)):

$$z_{0j} = \begin{cases} \max_i z_{ij} \; for \; stimulants \\ \min_i z_{ij} \; for \; destimulants \end{cases}, \tag{2}$$

$$z_{\_0j} = \begin{cases} \min_i z_{ij} \; for \; stimulants \\ \max_i z_{ij} \; for \; destimulants \end{cases}, \tag{3}$$

where $z_{0j}$ is the pattern and $z\_{0j}$ is the anti-pattern.

(3)　Euclidian distances (Equation (4)):

$$d_{i0} = \sqrt{\sum_{j=1}^{m}\left(z_{ij} - z_{0j}\right)^2} \quad (i = 1, \ldots, n).$$

(4)

Measure of development (Equation (5)):

$$m_i = 1 - \frac{d_{i0}}{d_0} \quad (i = 1, \ldots, n),$$

(5)

where $d_0$ is the distance between the pattern and anti-pattern of development determined, based on Equation (6):

$$d_0 = \sqrt{\sum_{j=1}^{m}\left(z_{ij} - z\_{0j}\right)^2} \quad (i = 1, \ldots, n)$$

(6)

The analysis was carried out for annual data covering the years 2004–2015. The choice of years for analysis was dictated by the availability and completeness of data in the Eurostat and OECD databases. The analyses were carried out for Poland and Hungary, as both these countries show similarities in the conditions of their economic development and initiatives undertaken in the field of waste logistics.

## 4. Results

The aim of the article is to propose a methodological framework to determine the dynamic relationship between variables described in detail in the previous chapter, in particular: balance and measure of development. The study of dynamic dependencies was preceded by a stationarity analysis carried out with the use of an ADF test. In addition, variables were transformed into natural logarithms (it was assumed that the prefix "l" before the variable designation means a transformation by the natural logarithm) and the test results are presented in Table 1.

**Table 1.** Results of the ADF–GLS test.

| Country | l_Balance | l_MR |
|---|---|---|
| Poland | tau = −2.30465 (0) p = 0.02046 c | tau = −1.94911 (0) p = 0.04906 c |
| Decision | I(0) | I(0) |
| Hungary | tau = −1.72622 (0) p = 0.07999 c | tau = −2.07688 (0) p = 0.03631 c |
| Decision | I(0) | I(0) |

Note: all variables in natural logs, lag length determined via MBIC in parentheses, ADF regression specification in deterministic part: c—constant.

The empirical level of significance of both variables for Poland is low. Hence, the hypotheses that the examined series are non-stationary at the level of significance of at least 0.05 should be rejected. Unfortunately, the results are not as explicit when it comes to variables estimated for Hungary. Therefore, the hypothesis about the non-stationarity of the measure of development variable can be rejected at the level of significance of at least 0.05, and the non-stationarity of the balance variable at the level of significance 0.08 In further studies, it was assumed that all variables are stationary. In order to determine the optimal order of variable delays in the vector-autoregressive model, the information criteria of Akaike (AIC), Schwarz-Bayesian (BIC) and Hannan and Quinn (HQC) were used. The results are presented in Table 2.

**Table 2.** Selection of the delay order for the VAR model.

| Delay order | AIC | BIC | HQC |
|:---:|:---:|:---:|:---:|
| | Estimates for Poland | | |
| 1 | 1.489077 | 1.706111 | 1.352267 |
| 2 | 1.417411 | 1.779134 | 1.189395 |
| 3 | 2.016195 | 2.522608 | 1.696973 |
| | Estimates for Hungary | | |
| 1 | −1.993673 | −1.776640 | −2.130483 |
| 2 | −1.585401 | −1.223678 | −1.813417 |
| 3 | −2.567420 | −2.061008 | −2.886642 |

The estimates for Poland show that both the Akaike and the Hannan and Quinn information criterion indicate a delay order of 2 (the smallest value of the information criterion). The Schwarz-Bayesian information criterion, on the other hand, indicates a delay order of 1. For Hungary all three information criteria indicate a delay order of 3. Next, the parameters of the vector-autoregressive model for Poland were estimated with delays order of 2, and for Hungary with a delays order of 3. It was confirmed that by adopting a delay order of 1 the whole set of explanatory variables did not significantly affect the explanatory variable in each of the equations.

The OLS method was used to estimate the VAR model parameters. The parameters estimated for Poland are shown in Tables 3 and 4.

**Table 3.** Parameters of Equation (1) of the VAR model. Explained variable: measure of development.

| Specification | Parameter | Standard Error | t-Student Statistic | *p*-Value |
|:---:|:---:|:---:|:---:|:---:|
| | Estimates for Poland | | | |
| const | −2.31427 | 1.76853 | −1.3086 | 0.23200 |
| l_Balance_1 | 0.0826229 | 0.158881 | 0.5200 | 0.61908 |
| l_Balance_2 | 0.0873833 | 0.162852 | 0.5366 | 0.60818 |
| l_MR_1 | 0.870263 | 0.285622 | 3.0469 | 0.01866 |
| l_MR_2 | −0.78259 | 0.263585 | −2.9690 | 0.02084 |
| | Estimates for Hungary | | | |
| const | 0.0177045 | 5.22404 | 0.0034 | 0.99746 |
| l_Balance_1 | −0.149381 | 0.446537 | −0.3345 | 0.75478 |
| l_Balance_2 | 0.311561 | 0.186388 | 1.6716 | 0.16993 |
| l_Balance_3 | −0.178976 | 0.105094 | −1.7030 | 0.16378 |
| l_MR_1 | 0.874016 | 0.370504 | 2.3590 | 0.07775 |
| l_MR_2 | 0.538032 | 0.95934 | 0.5608 | 0.60484 |
| l_MR_3 | −0.597734 | 0.507139 | −1.1786 | 0.30387 |

For Poland, there were explanatory variables with a statistically significant influence on the explained variable in Equation (1). For Hungary statistically significant variables are observed both in Equations (1) and (2). The results of Fischer-Snedecor, Ljung-Box and Doornik-Hansen tests confirmed the positive verification for Equation (1) for Poland and for both equations for Hungary (Table 5).

Due to the positive verification of the model defined by Equation (1) for Poland, the Granger causality was examined assuming a null hypothesis that the changes taking place on the secondary raw material market are not the cause (in the Granger sense) of changes in the environmental effects of the activities realized in the economy. In the case of Hungary, the Granger causality was checked for both variables. Values of the test in the form of Wald's statistics are presented in Table 6.

**Table 4.** Parameters of Equation (2) of the VAR model. Explained variable: balance.

| Specification | Parameter | Standard Error | t-Student Statistic | *p*-Value |
|---|---|---|---|---|
| | Estimates for Poland | | | |
| const | 6.74652 | 4.09701 | 1.6467 | 0.14361 |
| l_Balance_1 | 0.496144 | 0.368066 | 1.3480 | 0.21966 |
| l_Balance_2 | −0.281025 | 0.377266 | −0.7449 | 0.48059 |
| l_MR_1 | 0.222421 | 0.661677 | 0.3361 | 0.74661 |
| l_MR_2 | −0.263695 | 0.610626 | −0.4318 | 0.67884 |
| | Estimates for Hungary | | | |
| const | 18.1906 | 3.77256 | 4.8218 | 0.00851 |
| l_Balance_1 | −0.461932 | 0.322469 | −1.4325 | 0.22528 |
| l_Balance_2 | −0.26569 | 0.134601 | −1.9739 | 0.11963 |
| l_Balance_3 | −0.113926 | 0.0758943 | −1.5011 | 0.20773 |
| l_MR_1 | 1.92841 | 0.267562 | 7.2074 | 0.00196 |
| l_MR_2 | −0.00463514 | 0.692792 | −0.0067 | 0.99498 |
| l_MR_3 | 0.559865 | 0.366233 | 1.5287 | 0.20106 |

**Table 5.** F Fischer-Snedecor, Q Ljung-Box and Doornik-Hansen statistics for particular equations of the VAR model.

| Specification | F | *p*-Value for F Test | Q (for a Second Order of Delays) | *p*-Value for Q Test | Doornik-Hansen Test Statistic | *p*-Value for Doornik-Hansen Test Statistic |
|---|---|---|---|---|---|---|
| | | | Estimates for Poland | | | |
| Equation (1) | 4.063342 | 0.051583 | 1.22442 | 0.542 | 0.704156 | 0.7032 |
| Equation (2) | 0.661360 | 0.638202 | 0.300986 | 0.86 | 3.01624 | 0.2213 |
| | | | Estimates for Hungary | | | |
| Equation (1) | 3.640573 | 0.115796 | 1.63545 | 0.651 | 3.28691 | 0.1933 |
| Equation (2) | 30.84015 | 0.002611 | 1.95095 | 0.583 | 3.33316 | 0.1889 |

**Table 6.** The results of the Wald test for the Granger causality.

| Tasted Relation | The Value of the Test Statistic | *p*-Value |
|---|---|---|
| | Model for Poland | |
| l_Balance → l_MR | 0.39786 | 0.6860 |
| l_MR → l_Balance | 0.10413 | 0.9025 |
| | Model for Hungary | |
| l_Balance → l_MR | 2.5867 | 0.1905 |
| l_MR → l_Balance | 23.74 | 0.0052 |

The test results indicate the following:

- there are no grounds for rejecting the hypothesis about the lack of Granger causality for the variables distinguished for Poland;
- the existence of a one-way dependency in the case of Hungary is that the changes in the environmental effects of waste logistics activities are the cause (in the Granger sense) of the changes occurring in the trade of renewable raw materials.

Investigating the mutual impact of changes in the individual variables may provide information on short-term changes in the pro-environmental effects of waste logistics caused by unexpected changes in the trade of renewable raw materials in the context of inflow and outflow of waste materials. Impulse responses show how the synthetic measure of development for the effects of pro-environmental activities initially reacts to the shock in the difference between the sale and purchase of waste materials

in the trade in recyclable raw materials within the European Union, and whether the impact of the shock is permanent. A similar analysis was carried out for changes in the trade of raw materials caused by changes in the synthetic measure for environmental effects. Figures 3–6 show the functions of the impulse response for highlighted variables.

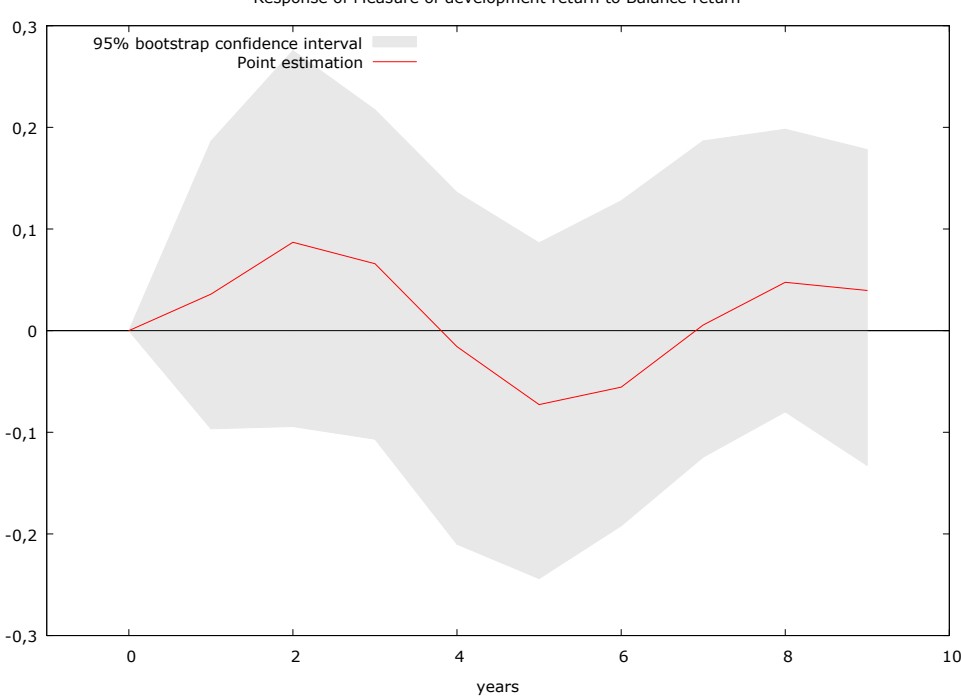

**Figure 3.** Impulse responses of measure of development to one standard deviation innovations for Poland.

The response of the measure of development to an impulse of one standard deviation—coming from the balance growth—is weak for both countries and heavily suppressed after six years for Hungary. For Poland, the balance increase reaction to the impulse caused by the measure of development is equally weak. This confirms the identified Granger causality relationships between the two variables for both countries. The situation is different when it comes to the balance increment reaction to the impulse caused by the measure of development for Hungary; it is stronger and lasts for another six years—however, after that period it is suppressed. It is worth emphasizing here that a quick response to the impulses appearing in the system and their quick suppression is indicative of the stability of the system. Thus, the response of the measure of development growth to the impulse from the balance growth side rather indicates a lack of stability in the case of Poland and Hungary. Stronger responses of the balance variable to the impulse from the side of pro-environmental effects may mean a higher degree of dependence between these variables in the analysed countries.

On the other hand, the decomposition of the forecast error variance indicates which part of the variability of the effects of pro-environmental activities in the area of waste logistics can be explained by the variability in the trade of raw materials, and vice versa. The decomposition of the variance of forecast error for the indicated variables is summarized in Tables 7 and 8.

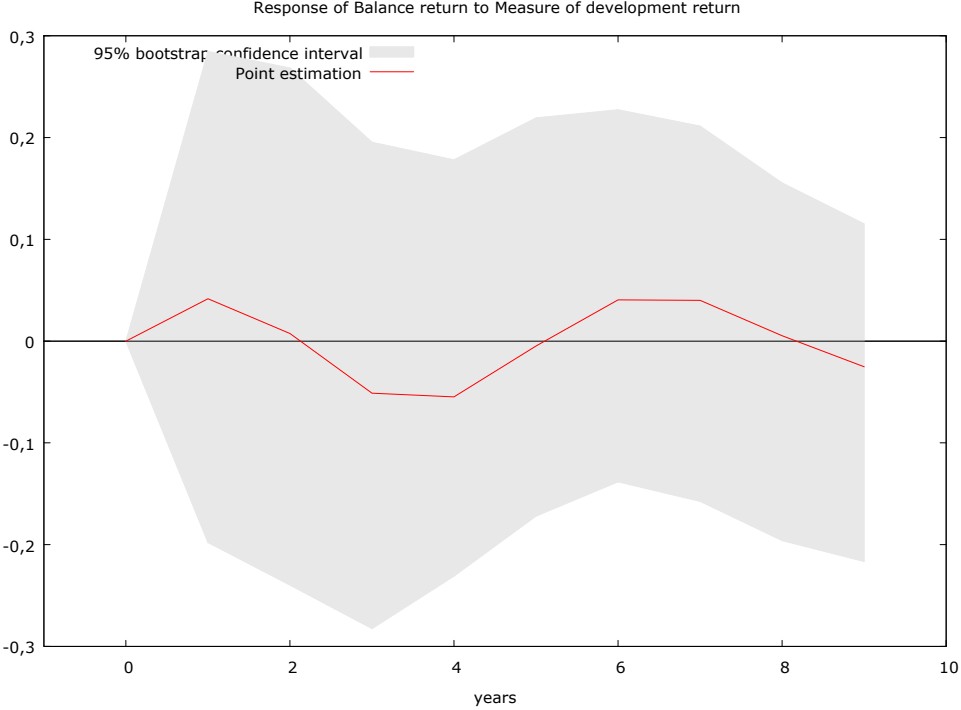

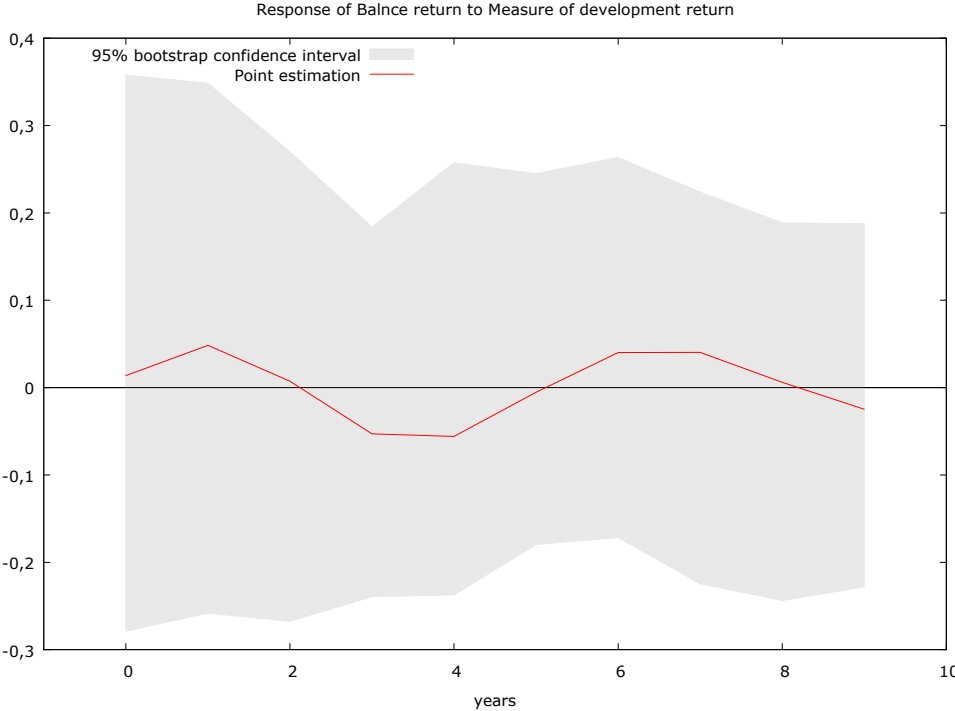

**Figure 4.** Impulse responses of balance to one standard deviation innovations for Poland.

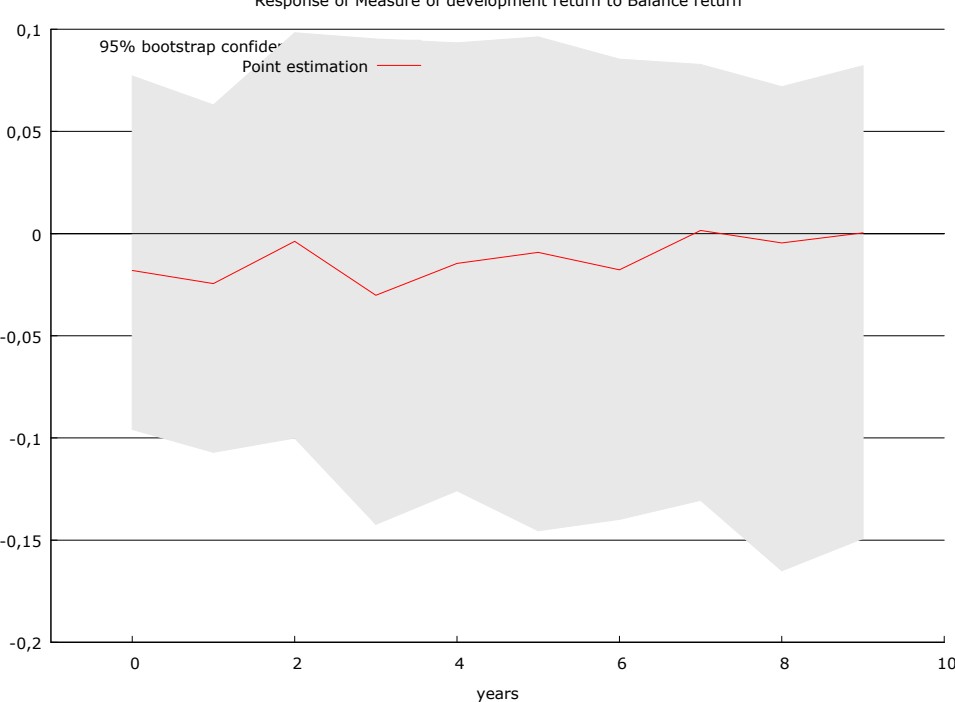

**Figure 5.** Impulse responses of measure of development to one standard deviation innovations for Hungary.

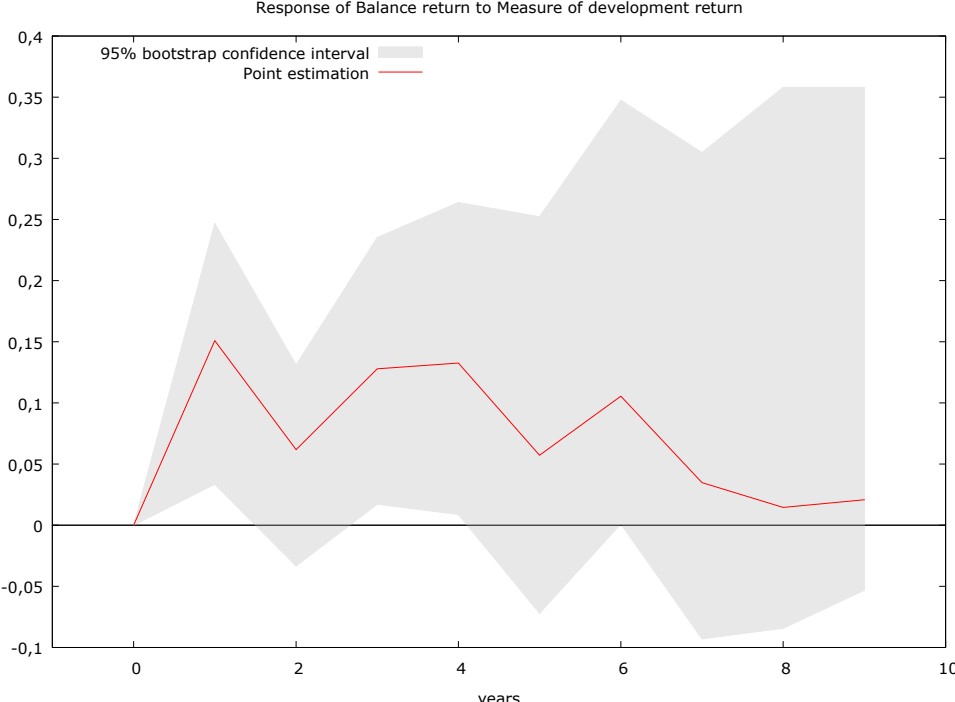

**Figure 6.** Impulse responses of balance to one standard deviation innovations for Hungary.

**Table 7.** The results of the variance decomposition of prediction error for measure of development.

| Horizon | Poland | | Hungary | |
|---|---|---|---|---|
| Impulse From | l_MR | l_Balance | l_MR | l_Balance |
| 1 | 100.0000 | 0.0000 | 94.9405 | 5.0595 |
| 2 | 97.9718 | 2.0282 | 92.1177 | 7.8823 |
| 3 | 87.5396 | 12.4604 | 94.7944 | 5.2056 |
| 4 | 85.4091 | 14.5909 | 93.4807 | 6.5193 |
| 5 | 86.9603 | 13.0397 | 93.5872 | 6.4128 |
| 6 | 82.7226 | 17.2774 | 94.1359 | 5.8641 |
| 7 | 81.3129 | 18.6871 | 93.6353 | 6.3647 |
| 8 | 82.1378 | 17.8622 | 93.6649 | 6.3351 |
| 9 | 80.6750 | 19.3250 | 93.6426 | 6.3574 |
| 10 | 79.9466 | 20.0534 | 93.6822 | 6.3178 |

**Table 8.** The results of the variance decomposition of prediction error for balance.

| Horizon | Poland | | Hungary | |
|---|---|---|---|---|
| Impulse From | l_Balance | l_MR | l_Balance | l_MR |
| 1 | 99.9003 | 0.0997 | 100.0000 | 0.0000 |
| 2 | 98.9297 | 1.0703 | 23.9246 | 76.0754 |
| 3 | 98.9077 | 1.0923 | 23.8263 | 76.1737 |
| 4 | 97.7786 | 2.2214 | 16.3368 | 83.6632 |
| 5 | 96.5506 | 3.4494 | 16.4729 | 83.5271 |
| 6 | 96.5461 | 3.4539 | 15.7630 | 84.2370 |
| 7 | 95.9260 | 4.0740 | 14.1121 | 85.8879 |
| 8 | 95.3050 | 4.6950 | 14.5920 | 85.4080 |
| 9 | 95.3007 | 4.6993 | 14.7806 | 85.2194 |
| 10 | 95.0775 | 4.9225 | 14.9714 | 85.0286 |

The results of the decomposition of the prediction error variance for the increments of the studied variables confirm the results of the Granger causality analysis and the impulse response function. The error in forecasting the increments of pro-environmental effects expressed by the synthetic measure of development depends mainly on their delayed values for both countries. A similar situation is observed in the trade of renewable raw materials. As the horizon of the forecast increases, the share of balance increments in the growth forecast error of the measure of development for both countries increases significantly. A similar pattern can also be observed for the share of the measure of development increments in the forecast error in the balance increments in Poland and Hungary.

## 5. Discussion

EU waste legislation gives a good basis towards building a circular economy model and developing a market for recyclable raw materials. The five-steps waste hierarchy places priority on the prevention of waste generation and preparing it for reuse. Recycling and recovery (including energy recovery), and finally disposal, i.e. landfill or thermal disposal are placed on subsequent positions.

The stringent targets for municipal waste recycling in the European Union assume:

– 55% recycling level by 2025;
– 60% recycling level by 2030;
– 65% recycling level until 2035.

Larger recycling rate is accompanied by landfill restrictions; by 2035, a maximum of 10% of municipal waste can be disposed of in this way. Segregation of bio-waste will become mandatory throughout the EU in 2023, and textiles will also have to be segregated from 2025.

According to the system categories approach characteristic for logistics, only a comprehensive way of dealing with a given problem can lead to the achievement of assumed goals. Therefore, what is

taken into consideration in an environmentally oriented logistics system is the interrelationship of flows connected with a given waste material and the change in its form caused by the applied ways of managing it. The environmental goals are treated as equivalent to the economic goals. Special attention is paid to the sozological goal determined as the minimization of resultant residues having a negative impact on the environment. This is how the environmentally oriented concept of logistics contributes to the minimization of environmental risk caused by the waste generated in the system. Waste management specialists and environmental experts must scan every technological and logistical possibilities in micro and macro level alike for sustainable development and minimized environmental pressures [65], as waste management is a sensitive area [66]. The ecological aspects can overwrite the well-meaning—but not uniquely optimized—regulations (laws, taxes, recommendations).

Beside waste minimization (zero waste strategy) we may develop the following areas in the EU [67]:

- recyclable raw materials market [68,69];
- waste logistics [70];
- closed waste treatment and recycling [71].

The conducted analysis should be deepened by including additional control variables in the analysis, e.g., the number of plants using secondary raw materials, the number of installations for processing secondary raw materials and energy and material consumption of installations using secondary raw materials in the production process. In addition, it is worth using econometric tools which allow for the identification of non-linear relationships between the analyzed endogenous variables. The conducted research enabled the analysis of dynamic relationships considering the export and import of waste materials and by-products to and from non-member countries of the European Union. Therefore, it would be worthwhile to carry out a similar analysis considering the entire trade of raw materials—inside and outside the EU.

## 6. Conclusions

Raw and recovered materials are becoming increasingly important in the management of materials. Minerals and organic raw materials often return to production systems as secondary raw materials. Due to the disposal of secondary raw materials, the increase in the demand for materials is slower than the increase in the demand for production in industrialised countries. Business entities strive to improve their production systems by eliminating all possible sources of waste, which is necessary, but also takes time, capital, effort and bold actions. Hence there is an ongoing search for tools that could enable the assessment of the effects of measures taken to avoid waste. Lack of solutions for treatment of waste means increased storage costs. The amount of waste deposited is defined as the difference between the total consumption of a given good and the quantity of the good intended for recycling. Therefore, the reduction in the amount of waste can be the result of an increase in the level of recycling and reduction of waste sources. For this reason, environmental protection instruments in the field of waste reduction should be directed at the product market and the secondary raw materials market.

Without a doubt the concept of waste logistics contributes to the development of trade of recyclable materials. The basic tasks connected with the implementation of waste logistics in enterprises include:

- reduction of resource consumption in the production phase;
- activities connected with the secondary flow;
- use of environmentally friendly technologies;
- actions to repair damages caused by company activity.

These activities contribute to sustainable development by creating closed material flows.

The conducted research revealed Granger causal relationships between the effects of the use of waste logistics activities and the development of trade of recyclable raw materials—particularly in Hungary. It was found that changes in the environmental effects of waste logistics activities are the

cause (in the Granger sense) of changes in the trade of renewable raw materials. This means that changes in the trade of recyclable raw materials can be better predicted if the correctly delayed values of the synthetic development measure for the environmental effects of waste logistics are considered. This pattern is also indicated by the reaction of the balance variable increment to the impulse from the measure of the development variable for Hungary, which is strong and lasts for the consecutive six years—although after that time it is suppressed.

The research carried out does not cover all analytical possibilities within the discussed topic. There is room for further research to verify the existence of dynamic relationships between the effects of waste logistics activities and the development of trade of recyclable materials in different countries. Poland and Hungary have similar economic conditions, which is why it would be reasonable to do a similar research including countries with differing economic, legal and political conditions. Both countries joined the European Union at the same time, and they are therefore characterized by an identical period of implementation of EU legislation in the field of waste management. The level of waste generated per capita is also similar. Poland and Hungary are characterized by a comparable level of recycling and landfill ratio as well. Additionally, it should be examined if similar relationships can be observed in the analysis of export to and import from countries both belonging and not belonging to the European Union. In addition, research can be extended to other European countries characterized by greater ecological awareness and responsibility of residents and longer practice in the field of sustainable waste management. It is highly recommended to extend the analysis period.

The problem of waste cannot be ignored, particularly as a continuous population growth is expected. The consequence will be a greater use of natural resources, and an increased pollution of the natural environment. For this reason, it is necessary to identify places, categories, quantities of waste and their impact on the environment. It is necessary to develop the assumptions of the waste management system both at the level of the enterprise, the local government unit and the state. In order to rationalise waste treatment, the planning, organizing and control tools should be used, as well as the entities responsible for waste management should be identified. The basis for such actions should be detailed statistics regarding waste, providing information on the level of waste generation and treatment.

**Author Contributions:** Conceptualization, A.M.-L. and P.M.; formal analysis, A.M.-L.; investigation, A.M.-L.; methodology, A.M.-L.; visualization, P.M.

**Funding:** This research received no external funding.

**Conflicts of Interest:** The authors declare no conflict of interest.

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
