# Peer review of "Sustainable Waste Logistics and the Development of Trade in Recyclable Raw Materials in Poland and Hungary"

_sustainability, doi:10.3390/su11154159_

Round 1

Reviewer 1 Report

The paper is engaged with a significant topic and can contribute both in academic literature and profession. It is a very serious work and fulfills in general its goal, based on an analytical description of a methodological framework for defining the interrelation between two very modern concepts.

The very well designed methodology is the strong attribute of this paper. However, I would note the following :

While the aim of the article is "to determine the degree of the dynamic impact of the effect of activities in the field of sustainable waste logistics on the development of trade in recyclable raw materials.." or "...to determine the strength of the interaction between these variables." , this is not adequately met in the Discussion and Conclusions paragraphs regarding the cases of the two countries. For instance, there is no discussion about the differences between the results in these countries. How are these explained ? Is that really the case in terms of the papar's goal ? : to analyze the degree of impact or possibly to suggest a methodological framework-tools in order to determine the interrelation between the waste logistics and trade in recyclable raw materials, taking into account the relevant factors?

It seems more like the aim is to describe a methodological framework that will lead to the aims mentioned by the author(s)!!

It is strongly recommended to change the research hypothesis as formulated, since anyway as mentioned in the conclusions "Without a doubt the concept of waste logistics contributes to the development of trade of recyclable materials." So therefore, the hypothesis is somehow expected to be confirmed by the analysis, as a very reasonable result, and in this sense it is not very value-adding; whereas the methodology developed is very interesting contributing significantly in the waste management research field.

The "similarity" of Poland and Hungary is not adequately documented. In this sense, some quantitative information is necessary. "Similar economic conditions" is vague and is not necessarily associated with the way of managing waste logistics and trade of recycling materials (i.e. mechanisms, laws, etc.).

Author Response

Thank you very much for all suggestions. As suggested in the article, the following changes were made:
1. The goal has been adapted for research.
2. The hypothesis has been changed - it refers to the applied methodology.
3. The similarities between Poland and Hungary were justified.

4. A linguistic correction was made.

According to the suggestions of other reviews the following changes were made in the article:

1.      In section 4 it has been clarified what data were used from 2004 and 2015.

2.      The part concerning waste logistics has been expanded. I would like to mention, however, that the discussion on the implementation and realization of logistic processes in the field of waste logistics was not the subject of the article.

3.      A figure showing the recycling rate was developed. The main methods of waste management in Poland and Hungary are indicated.

4.      The figure presenting impulse responses has been improved and the discussion of the figure has been extended.

5.      The discussion has been extended to the EU guidelines on waste treatment.

6.      The summary has been extended to include proposals for action on waste growth due to population growth. It has been shown how the study can be extended.

Reviewer 2 Report

The paper deals with an interesting topic but is extremely hard to read and I had to read multiple times in order to not get lost in the narrative.

THe paper can be improved if the authors avoid using extremely large sentences (up to 5 lines at times) that make it a very difficult to read. Please split long sentences as even the abstract feels hard to read.

It would also benefit the manuscript if it was proofread by a native speaker. For example even the first sentence contains an error (An European citizen, should be A European citizen).

I would expect a short summary of the guidelines of EU regarding waste treatment. Is landfill better than incineration? What are the main modes in Hungary and Poland that you are examining in your case study? In section 4 you present your analysis results based on the data, but you do not show/describe what data you used from 2004 and 2015. Please elaborate.

Also you speak about waste logistics but not much is explained. How are waste transported? Trucks? Rail? waterborne? See a discussion from:

Zis, T., Bell, M. G. H., Tolis, A., & Aravossis, K. (2013). Economic evaluation of alternative options for municipal solid waste management in remote locations. Waste and Biomass Valorization, 4(2), 287-296.

Lines 29-31 you provide some statistics. Please add a barchart for each member state.

Improve Figure 2 as it is hard to read, and please expand the discussion section of this Figure.

Improve your conclusions section. How are your results transferable? What will happen with the population growth and the increase in waste generation? How can we deal with this issue? I think a short comment on these items would improve the paper.

Author Response

Thank you very much for all suggestions. As suggested in the article, the following changes were made:

1.      In section 4 it has been clarified what data were used from 2004 and 2015.

2.      The part concerning waste logistics has been expanded. I would like to mention, however, that the discussion on the implementation and realization of logistic processes in the field of waste logistics was not the subject of the article.

3.      A figure showing the recycling rate was developed. The main methods of waste management in Poland and Hungary are indicated.

4.      The figure presenting impulse responses has been improved and the discussion of the figure has been extended.

5.      The discussion has been extended to the EU guidelines on waste treatment.

6.      The summary has been extended to include proposals for action on waste growth due to population growth. It has been shown how the study can be extended.

7.      A linguistic correction was made.

According to the suggestions of other reviews the following changes were made in the article:

1. The goal has been adapted for research.
2. The hypothesis has been changed - it refers to the applied methodology.
3. The similarities between Poland and Hungary were justified.

4. A linguistic correction was made.

Reviewer 3 Report

The present paper analyzes a highly topical subject. The processing method is innovative and the choice of methods used can be highly appreciated. The results of the analyzes lead to conclusions that the author adequately commented on.

Author Response

Thank you very much for the review. Some changes were made in the article due to the suggestions of other reviews:

1.      The goal has been adapted for research.

2.      The hypothesis has been changed - it refers to the applied methodology.

3.      The similarities between Poland and Hungary were justified.

4.      In section 4 it has been clarified what data were used from 2004 and 2015.

5.      The part concerning waste logistics has been expanded. I would like to mention, however, that the discussion on the implementation and realization of logistic processes in the field of waste logistics was not the subject of the article.

6.      A figure showing the recycling rate was developed. The main methods of waste management in Poland and Hungary are indicated.

7.      The figure presenting impulse responses has been improved and the discussion of the figure has been extended.

8.      The discussion has been extended to the EU guidelines on waste treatment.

9.      The summary has been extended to include proposals for action on waste growth due to population growth. It has been shown how the study can be extended.

A linguistic correction was made.

Reviewer 4 Report

The paper is very well structured and deals with the selected topics in a professional way.

The authors use modern mathematical methods for the interpretation of the results.

The outcome of the paper is very useful for Central European countries.

Some EN polsihing  is recommended: re-use, or reuse,  re-cycle or recycle.

This is to recommend the paper after minor revision.

Author Response

(The authors gave the same response as above.)

Round 2

Reviewer 2 Report

    The authors have not adequately addressed my comments, the review remains weak, I therefore regretablly have to propose rejection.